# Nutritional Quality of Meat from Barren Merino Ewes in Comparison to Meat from Traditional Lambs

**DOI:** 10.3390/ani13172756

**Published:** 2023-08-30

**Authors:** Begoña Panea, Guillermo Ripoll, María J. Alcalde

**Affiliations:** 1Animal Science Department, Centro de Investigación y Tecnología Agroalimentaria de Aragon (CITA), Avda. Montañana 930, 50059 Zaragoza, Spain; bpanea@cita-aragon.es; 2Agrifood Institute of Aragon-IA2 (CITA-Zaragoza University), Avda. Miguel Servet 177, 50013 Zaragoza, Spain; 3Department of Agronomy, Universidad de Sevilla, Ctra. Utrera km. 1, 41013 Seville, Spain; aldea@us.es

**Keywords:** barren ewe, lamb, sex effect, nutritional value, meat quality

## Abstract

**Simple Summary:**

Barren ewes represent approximately 2% of the census of the herd, which constitutes a major loss in revenue for the farmer. It would, therefore, be an interesting market strategy to revalue the meat of these barren ewes. We studied the nutritional quality of barren ewes’ meat in comparison to traditional lambs’ meat by studying 10 barren ewes, 10 male lambs, and 10 female lambs from the Merino breed. There was no effect of animal type (males, females, and ewes) on pH, and differences in subcutaneous fat color, lipid oxidation, and texture were irrelevant from a practical point of view. The tissue composition, in the three groups of animals, reflected a high percentage of saleable meat, with no penalty for intensive fattening in any of the groups. The ewes’ meat presented a higher percentage of moisture, collagen, ash, calcium, iron, α-tocopherol, and retinol than the lambs’ meat. In addition, it had a higher content of DHA and CLA and lower values for the ratio n6/n3, which is beneficial for health, although it also contained more fat, saturated fatty acids, and cholesterol than lambs’ meat. Ewes’ meat therefore has a good nutritional composition and can be considered an attractive product for consumers.

**Abstract:**

In Spain, lamb consumption has decreased over the last few years. To increase farmers’ income, we studied the nutritional quality of the barren ewes’ meat in comparison to traditional lambs’ meat with 10 barren ewes, 10 male lambs, and 10 female lambs from the Merino breed. We measured the subcutaneous fat, muscle color, and carcass tissue composition, as well as proximal composition, mineral, tocopherol, retinol, lutein, and cholesterol contents, and the TPA texture profile, fatty acid profile, and lipid oxidation of the meat. There was no effect of the animal type (males, females, and ewes) on the pH, and the differences in the subcutaneous fat color, lipid oxidation, and texture were irrelevant from a practical point of view. The tissue composition in the three groups of animals reflected a high percentage of saleable meat, with no penalty incurred for intensive fattening in any of the groups. The ewes’ meat presented a higher percentage of moisture, collagen, ash, calcium, iron, α-tocopherol, and retinol than the lambs’ meat. In addition, it had higher content of DHA and CLA and lower values for the ratio n6/n3, which is beneficial for health, although it also contained more fat, saturated fatty acids, and cholesterol than the lambs’ meat.

## 1. Introduction

Spain is one of the largest producers of sheep meat in the EU-27. In Spain, most sheep farms are located in marginal areas, which are not used for other activities and are often classified as high nature value farmland [1]. These livestock systems are multifunctional [2] and provide meat and meat products with extrinsic characteristics that are becoming increasingly important for the consumer [3], such as the local origin or environmentally friendly production [4]. Despite this, lamb consumption has fallen in recent years, so it is essential to seek strategies to increase farmers’ income, since their work is essential to fixing the sheep population in sparsely populated areas. In this context, the future of farms necessarily involves taking advantage of the extrinsic quality characteristics, linking production to the territory through the exploitation of native breeds and obtaining differentiated, high-quality products [5].

In Spain, the meat production of some native ovine breeds is almost exclusively limited to a single product, often endorsed by a quality label. In the case of the Merino breed, the P.G.I. “Cordero de Extremadura” protects the meat of lambs that are slaughtered before 100 days and whose weight is less than 16 kg for males and 14 kg for females. This system responds to the requirements of the majority of Spanish consumers, who prefer the meat of younger animals [6]. However, with new market trends encouraging product diversification as a means of increasing market share, it is advisable to value the meat of other types of animals, such as barren ewes. Barren ewes are classified as those that are permanently sterile or that had their last calving more than two years previously. Barren ewes represent approximately 2% of the census of the herd, which constitutes a major financial loss for the farmer, since they consume resources but do not produce. Their only possible use is therefore culinary, and their meat has traditionally been highly appreciated among farmers. For this reason, it would be an interesting market strategy to revalue barren ewes’ meat. Taking into account the fact that the age and sex of an animal greatly influence the quality of its meat, including nutritional characteristics [7], and that consumers are becoming increasingly aware of the relationship between nutrition and health, consumers’ purchasing decision can be helped by determining the nutritional characteristics of barren ewes’ meat [8]. Therefore, this study was carried out at the request of the National Association of Merino Cattle Breeders to evaluate the nutritional quality of the meat of barren ewes in comparison to traditional lambs’ meat.

## 2. Material and Methods

### 2.1. Animal Rearing and Slaughtering

This study was conducted in southwest Spain, where all of the animals were reared on commercial farms. We worked with the following types of animals:-Barren ewes (10 animals, referred to hereinafter as “ewes”): The animals came from a single farm, located in Trujillo, where they were raised under an extensive management system with a reproductive system of three births in two years. It is a transhumant flock, and the sheep usually spend the winter in Trujillo and go up to the mountains of Leon in summer to pasture. Ewes that have not become pregnant one year after being incorporated into the flock as breeders are usually eliminated from the flock;-Lambs: The animals came from 12 farms located in the provinces of Cáceres and Badajoz. A total of 20 lambs from single parturitions (10 females and 10 males) were reared on their respective farms until weaning. After weaning, they were moved to the Animal Selection and Reproduction Centre of Extremadura (Badajoz, Spain, http://www.gobex.es/con03/censyra, accessed on 1 June 2023), where they were fed with a mixed ration *ad libitum*. The chemical composition of the mixed rations for the lambs was as follows: 91.1% dry matter, 15.9% protein, 3.8% fat, 7.6% ash, 16.5% NFD, 6.4% ADF, and 0.1% ADL. For the duration of the trial, all animals were weighed weekly, and the lambs were slaughtered when they reached the live weight of 22–25 kg (approximately 115 days of age).

All animals were slaughtered at a slaughterhouse in Agudo (Spain, 38°58′45″ N 4°52′23″ W), following European regulations [9]. The hot carcass weight (HCW) was recorded, and the carcasses were hung by the calcaneus tendon and kept at 4 °C/24 h until fat color measurements were taken.

### 2.2. Subcutaneous Fat Color and Tissue Composition

The color of the subcutaneous fat of all carcasses was measured at the 8th rib level with a Minolta CM-600d Spectrophotometer (Konica Minolta Holdings, Inc., Osaka, Japan) in CIEL*a*b* space (CIE, Cambridge, UK, 1986) with the specular component including 0% UV, observer angles of 10 and 0, and white calibration. The measurement area was 8 mm in diameter, and the illuminant used was D65. The spectrophotometer was rotated on the horizontal plane before each reading, and the mean of three readings was used for analysis. The lightness (L*), redness (a*), and yellowness (b*) indices were recorded using SpectraMagic NX v. 3.31 software (Minolta Co., Ltd., Osaka, Japan), and the hue angle $\left(h_{\mathrm{ab}}=\tan^{-1}(\left(b^{*}\right)/\left(a^{*}\right))\cdot 180{}^{\circ}/\Pi \right)$ and chroma ($C_{\mathrm{ab}}^{*}=\sqrt{{\left(a^{*}\right)}^{2}+{\left(b^{*}\right)}^{2}}$) were calculated.

After the color measurement, the left forelimb was separated from the carcass [10], vacuum-packed, and stored at −20 °C until sampling. After thawing overnight at 4 °C, the forelimb was weighed and dissected into muscle, intermuscular fat, subcutaneous fat, bone, and others (major blood vessels, ligaments, tendons, and thick connective tissue sheets associated with muscle). The tissue composition of the forelimb was expressed as a percentage of the thawed forelimb weight, following Panea, Ripoll, Albertí, Joy, and Teixeira [10].

### 2.3. Meat Quality Sampling and Procedures

The left *longissimus thoracis et lumborum* muscle and the left *triceps brachii* muscle were removed and used for the meat quality measurements.

The pH was measured at 24 h post mortem on the cranial side of the *longissimus thoracis et lumborum* muscle with a Crison pH meter pH/mv-506 m, and the following samples were obtained, vacuum-packed, and frozen at −20 °C until the analysis.

From longissimus thoracis et lumborum:-Proximal composition, using a FoodScan 2^®^ NIRS (Foss Iberia, S.A., Barcelona, Spain); Percentages of moisture, protein, intramuscular fat, saturated fat, collagen, and ash were recorded;-Measurement of mineral content. Samples were previously lyophilized using a VirTis Genesis machine (SP Scientific Products, Warminster, PA, USA), and the mineral content was measured using an i CAP PRO XP Duo ICP Plasma Atomic Emission Spectrometer (ICP-OES, Thermo Fisher Scientific Inc., Madrid, Spain), following an internal procedure of the Analysis Service of the University of Zaragoza, Zaragoza, Spain), with duplicate measurements taken;-Tocopherol, retinol, lutein, and cholesterol contents were measured, following Bertolín et al. [11], by means of a chromatographic system using an ACQUITY UPLC H-Class liquid chromatograph (Waters, Milford, MA, USA) equipped with a silica-based bonded phase column (Acquity UPLC HSS T3, 1.8 μm × 2.1 mm × 150 mm column (Waters, Milford, MA, USA), an absorbance detector (Acquity UPLC Photodiode Array PDA eλ Detector, Waters), and a fluorescence detector (2475 Multi λ Fluorescence Detector, Waters). The UHPLC system was controlled using Empower 3 software. Duplicate measurements were taken;-TPA texture profile was measured, following Ripoll et al. [12], using an Instron machine model 5543 (Instron Limited, Cerdanyola, Spain). A minimum of 5 probes were obtained by sample, measuring hardness (maximum force exerted in the first compression cycle, in N) and adhesiveness (area of negative force obtained after the first compression, representing the force needed to separate the compression plunger from the food, in MJ);-Muscle color, using a Minolta CM-600d spectrophotometer (Konica Minolta, Osaka, Japan) in the same way as described for fat color.

From the *triceps brachii*:
-To separate the fatty acid profile from its quantitative percentage, an Agilent 6890N model gas chromatograph (Agilent Technologies, Santa Barbara, CA, USA) was used, equipped with a flame ionization detector. An HP-88 capillary column 100 m long, with an internal diameter of 0.25 mm and a phase thickness of 0.2 µm, was used in the stationary phase. H_2_ was used as the carrier gas, with a flow of 2 mL min^−1^. The initial temperature of the oven was 100 °C, which was then increased by 3 °C per minute until it reached 158 °C; it was then increased further by 1.5 °C up to 190 °C, and this temperature was maintained for 15 min. Next, the temperature was raised to 200 °C with an increase of 2 °C per minute. Finally, a rapid rise of 10 °C per minute was made until it reached 240 °C, and this temperature was maintained for 10 min. The injector temperature was programmed at 300 °C and the detector temperature at 320 °C. The split mode was used for the injection. The identification of the fatty acids was carried out by comparing their retention times with the retention times in a mixture of patterns [13]. Duplicate measurements were taken;-Lipid oxidation was measured by determining the TBARS index. Once thawed, the samples were cut into two pieces: one was used to determine the lipid oxidation level at day 1, and the other one was kept in the dark at 4 °C for 7 days and used to determine lipid oxidation at 7 days of aging. Lipid oxidation [14,15] was expressed as mg of malondialdehyde (MDA)/kg fresh meat. Duplicate measurements were taken.

### 2.4. Statistics

All the statistics were calculated using the XLSTAT statistical package v.3.05 (Addinsoft, New York, NY, USA). Means were calculated for all of the variables studied. An ANOVA was performed with the type of animal (male, female, and ewe) as a fixed effect. The differences among the samples were measured with a Tukey test and a significance of *p* < 0.05.

## 3. Results and Discussion

### 3.1. Subcutaneous Fat Color

The values found for the subcutaneous fat color are shown in Table 1. The values are comparable to those collected by other authors in Merino breed animals of similar weights [16,17]. Differences among types of animals were found only for luminosity (L*), which was higher in ewes than in lambs, which could be related to the fatty acid profile. Although, in general, the literature concurs that the color of subcutaneous fat is mainly due to animal feeding, tending to be more yellow in grass-based diets [18], this effect was not found in our study. In addition, the white color of the body fat of sheep and goats is thought to be mainly due to the extremely low concentration of carotenoids in these animals [19].

From a practical point of view, the small differences found in this paper are of no practical interest.

### 3.2. Tissue Composition

Table 2 shows the tissue composition of the shoulders. This piece is representative of the tissue composition of the entire carcass [10]. The current data are consistent with those reported by other authors in several Spanish breeds [20,21].

The weight of the ewes’ shoulders was greater than that of the lambs, as expected. Differences among types of animals were found only for the percentage of muscle, with the ewes presenting a percentage of muscle similar to that of male lambs. Although some authors [22] have reported that after dissecting lambs’ shoulders, females were found to have higher fat percentages and males higher bone percentages, in the current study no differences in the lambs were found by sex.

### 3.3. pH and Proximal Composition

Table 3 shows the pH and proximal composition data of the meat depending on the type of animal. No differences in pH were found depending on the type of animal, in line with several authors [7,23]. The current values are usual for sheep meat [7,24,25,26], indicating that there was no effect of the pH on the rest of the measured parameters.

The values found for the proximal composition agreed with those described by other authors for animals with similar characteristics [7,24,26].

The ewes’ meat presented a higher percentage of moisture, fat, saturated fats, collagen, and ash than that of the lambs’ in which no influence of sex was found. This lack of differences between males and females in fat content in young animals has been described by other authors [24,27]. However, we did expect differences in fat content due to the animals’ weight [7,28], with a major difference in collagen content depending on their age. There is little literature on the collagen content of sheep meat, but the results we found for lambs coincide with those described by Baila, Lobon, Blanco, Casasus, Ripoll, and Joy [25] for suckling lambs (0.77%), and the data from the ewes agreed with those described by Tschirhart-Hoelscher et al. [29] for animals with a carcass weight of approximately 30 kg (2.6–2.9 mg/g fresh meat) or those described by Ramírez-Zamudio et al. [30] for ewes (2.05–2.3 mg/g fresh meat).

Regardless of the differences due to the animals’ sex or weight, it is clear that sheep meat is an excellent source of protein and that the percentages of saturated fat are moderate. The EFSA recommends that the percentage of saturated fat should be <10% of the total energy consumed, which for a diet of 2000 Kcal/day is approximately 16–20 g of saturated fat/day. Here, a standard 200 g ration of lamb would provide only 2 g of saturated fat in the case of lambs and 4.6 g of saturated fat in the case of ewes.

### 3.4. Mineral Content

The mineral content of each type of meat is shown in Table 4. Minerals are essential inorganic elements for the body and cannot be synthesized, so they must be included in the diet. They intervene in tissue structure (calcium, phosphorus, and magnesium), control the composition of body fluids (sodium, chlorine, potassium, magnesium, and phosphorus), and are part of enzymes and other proteins involved in metabolism.

The values here are similar to those described by other authors in animals with similar characteristics [31,32,33].

Except for Zn, there are differences in the mineral content between ewes’ meat and that of lambs. Ewes’ meat contains higher amounts of calcium and iron and lower amounts of the other minerals than lambs’ meat. Again, there is no influence of sex among lambs. Novoselec, Salavardic, Samac, Ronta, Steiner, Sicaja, and Antunovic [18], working with Merino lambs between 95 and 125 days of age, pointed out that the mineral content increases with the animal’s age, especially magnesium and iron, which could be related to the proportion of each type of muscle fiber, depending on the age [34]. In the present work, this effect was found only for calcium, which is associated with bone metabolism, and for iron, which is due to the meat of adult animals having a greater amount of myoglobin than that of lambs [35]. On the other hand, Holman et al. [32] reported that the addition of lucerne to the animals’ diet resulted in increased iron and phosphorus content, which reinforces the current results for iron in ewes which were fed mainly on forage.

In humans, the nutritional daily recommendations for minerals [36] in adults are as follows: calcium (1000 mg), iron (8–18 mg), magnesium (300–400 mg), phosphorus (1000 mg), potassium (2800–3800 mg), sodium (450–900 mg), and zinc (8–14 mg). A 200 g serving of Merino meat would provide 12–22 mg of calcium, 4–7 mg of iron, 690–800 mg of potassium, 50–57 mg of magnesium, 90–120 mg of sodium, 490–460 mg of phosphorus, and 5–6 mg of zinc.

### 3.5. Content in Tocopherol, Retinol, Lutein, and Cholesterol

Vitamin E is composed of a set of eight stereoisomers. It cannot be synthesized by the animals, so it must be ingested with the diet [35]. Of these, α-tocopherol is the most active as a vitamin and γ-tocopherol as an antioxidant. When vitamin E levels are high, vitamin A absorption decreases, as they compete for the same absorption mechanisms [19,37]. In addition, carotenoids are widespread isoprenoid secondary metabolites, some of which can be converted into vitamin A in animals. However, animals cannot synthesize carotenoids de novo and rely on the diet as a source. These vitamins are therefore significant from a nutritional point of view, and the role of fat-soluble vitamins in the nutritional and sensory properties of foods has been pointed out [38].

A study carried out by Álvarez, Meléndez-Martínez, Vicario, and Alcalde [37] in lambs showed that carotenoids were not detected in any fat sample. However, this is in disagreement with Yang, Larsen, and Tume [19], who reported that lutein was present in the adipose tissue of sheep, albeit in very low quantities, as found in our work.

On the other hand, Alvarez [37] reported that both retinol and α-tocopherol were detected in the fat samples of the lambs studied. The levels of α-tocopherol were higher than those of retinol in the groups, which was expected, as adipose tissue is one of the main storage sites of tocopherol, while for retinol it is the liver [39,40].

An effect of animal type was found for all of the variables (Table 5), except for γ-tocopherol. Because these compounds are fat soluble and are deposited in intramuscular fat, which is higher in ewes, the ewes’ meat contained more α-tocopherol, retinol, and cholesterol than that of the lambs’, in which there are no differences. On the other hand, the meat of the females presented a higher content of δ-tocopherol than the meat of males or ewes, which showed no differences due to sex.

No lutein was detected in the male lambs’ meat and, as expected, the content was much higher in the ewes’ meat than in the female lambs’ meat. There is little literature with data on vitamins and cholesterol in Merino lambs. However, Campo, Silva, Guerrero, Castro, Olleta, Martin, Fernández, and López [33], working with lambs of several Spanish breeds (including Merino), found α-tocopherol values of between 0.27 mg and 0.31 mg/100 g edible portion (muscle + visible fat), which is clearly lower than those found in the present experiment. Despite this, those authors found cholesterol values of 0.65–0.68 mg/100 g edible portion, which is in line with those found in our experiment.

### 3.6. Fatty Acid Profile

The results (shown in Table 6) agree with those found by other authors for the Merino breed [41,42,43,44]. We found an effect of the type of animal in 23 of the 41 fatty acids detected and, in general, the content of almost all of the fatty acids was higher in ewes’ meat than in lambs’ meat, probably due to the higher fat content of the former [45]. Numerous studies have indicated that increased intramuscular fat content is associated with an increase in total saturated fatty acids and a decrease in polyunsaturated fatty acids [46]. However, in our study, we found no differences among animal types for these values.

The only differences found were among lambs in C18:1n9c, with higher values in the meat of males than females. This low effect of sex in the young animals is in line with the results of other authors [23,44].

The ewes’ meat had a higher content of DHA and CLA fatty acids than that of the lambs’, which is considered to have health benefits for humans [43]. It has been described that the CLA content depends fundamentally on the diet, so that grass-based diets favor higher levels of CLA in meat [47], which could explain the results found in the ewes’ meat.

On the other hand, the ewes’ meat presented a lower percentage of n-6 fatty acids and a higher percentage of n-3 fatty acids than the lambs’ meat, resulting in lower values for the n6/n3 ratio. Our results partially coincide with the conclusions of Santos-Silva et al. [48], in Merino Branco with a live weight of 24 or 29 kg, who describe how the proportion of n-6 and n-3 acids decreases with an increased slaughter weight, although they found no effect of slaughter weight for the ratio n-6/n-3. Values greater than seven in the ratio n6/n3 are typical of animals raised on cereals [33], which would account for the ratio found for lambs.

The n-6 and n-3 fatty acids are key for maintaining the structure of cell membranes, facilitating the absorption of fat-soluble vitamins, regulating cholesterol metabolism, and controlling homeostasis. Despite the fact that numerous intervention studies have shown that a diet rich in n-3 fatty acids reduces coronary mortality and sudden cardiac death and that it is important to have an adequate n6/n3 ratio, the maximum levels of intake for n-6 and n-3 fatty acids have not yet been established. There is also no agreement on the best ratio of DHA and EPA. Most guidelines recommend 20–35% of total energy in the form of fats, divided into 7–10% saturated fat, 20% monounsaturated, and 6–10% polyunsaturated. These recommendations also state that the correct intake of n-3 fatty acids is 0.5–2 g/day with an upper limit of 3 g/day, while that of n-6 fatty acids is between 2.5 and 10 g/day, with a recommended n-6/n-3 ratio of 5:1 [49]. The ewes’ meat in our trial complies with this ratio, although the lambs’ meat exceeds the recommended limit. On the other hand, the recommended content of EPA and DHA is at least 500 mg daily [49], and one 200 g serving of Merino meat could provide between 30 and 75 mg of EPA+DHA.

The meat of adult animals often has a bad reputation because it is supposed to have an unhealthy lipid profile; however, our data have shown that, beyond the fact that there may be differences in any specific acid, probably due to differences in the animals’ handling, ewes’ meat can be considered just as healthy as lambs’ meat.

### 3.7. Lipid Oxidation (TBAR)

Lipid oxidation (Figure 1) increased over the storage time, as expected [50,51]. There were no differences among types on the first day, but, surprisingly, on day 7 the lambs’ meat presented higher oxidation values than that of the ewes’, although no differences in unsaturated fatty acids sums were detected. However, both lambs’ and ewes’ meat presented very low lipid oxidation values at 7 days, which was much lower than those described by other authors in the meat of the Merino breed aged 7 days [43,50,51].

### 3.8. Texture Profile

In the TPA (texture profile analysis) assay, a plunger compresses a sample uniaxially and twice consecutively to simulate jaw movement during chewing. Thus, the analysis of the curve obtained allows us to calculate several texture parameters correlated with the sensory evaluation. Hardness is defined as the maximum force exerted in the first compression cycle, while adhesiveness represents the work necessary to separate the plunger from the food and represent the sticky mouthfeel. Springiness is defined as the height at which food can recover between the first and the second bites, and compression is defined as the ratio between the force needed for the first and the second bites [52].

In our study, slight differences were found for cohesiveness and springiness, while no differences were found either for hardness or for adhesiveness (Table 7), which coincides with the results of other authors [28] and indicates that ewes’ meat is similar, in terms of texture, to that of lambs. The values found for hardness, however, are much lower than those reported in animals of similar characteristics by other authors [17,42,53,54]. Martinez-Cerezo, Sanudo, Panea, Medel, Delfa, Sierra, Beltran, Cepero, and Olleta [7], in a study that compared the Merino with other breeds and studied the effect of slaughter weight (20 kg or 30 kg live weight), found that there were no differences in texture depending on the animal’s weight and that Merino meat was more tender than that of the other breeds, which may be due to the high solubility of collagen in this breed. This would explain why ewes’ meat, which has a higher amount of collagen than that of lambs, is not, however, harder than lambs’ meat.

### 3.9. Meat Color

The results for meat color are shown in Table 8. The values match with those described by other authors for the Merino breed in animals of similar weights [17,22,23,44,54]. The ewes’ meat registered lower values of luminosity (L*) and tone (*h*_ab_) and higher values of red index (a*) than the lambs’ meat. The color of the meat is mostly due to myoglobin and its iron atom, which explains why the ewes’ meat is redder than that of the lambs’, since the content of myoglobin and iron increases with the age of the animal, and the ewes’ meat has a higher iron content than that of lambs’ [35], as shown in Table 6. The absence of effect of sex on meat color when the animals are young has been described by several authors [22,23].

## 4. Conclusions

There was no effect of animal type (males, females, and ewes) on pH, and differences in subcutaneous fat color, lipid oxidation, and texture were irrelevant from a practical point of view. The tissue composition, in the three groups of animals, reflects a high percentage of saleable meat, without there being a penalty for the intensive fattening in any of the groups. The ewes’ meat presented a higher percentage of moisture, collagen, ash, calcium, iron, α-tocopherol, and retinol than the lambs’ meat. In addition, it had a higher content of DHA and CLA, as well as lower values for the ratio n6/n3, which is beneficial for health, although it also contained more fat, saturated fat, and cholesterol than the lambs’ meat. In view of the results obtained, we can consider that ewes’ meat, despite there being some differences between male and female lambs, has a good nutritional composition and adequate physicochemical characteristics and can be an attractive product for marketing.

## Figures and Tables

**Figure 1 animals-13-02756-f001:**
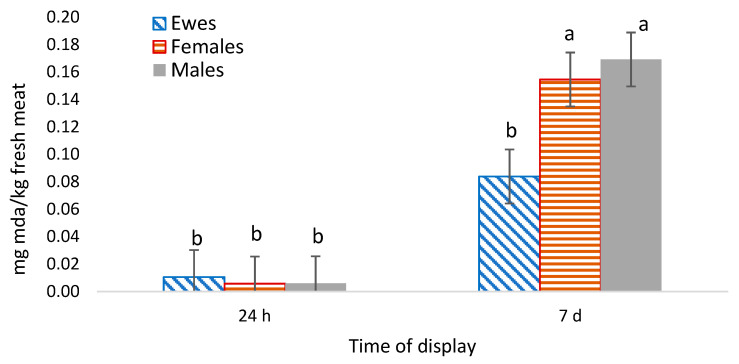
Evolution of lipid oxidation values (TBAR) over storage time, depending on the type of animal (means and standard error bars). a, b—Differences among batches (*p* < 0.05).

**Table 1 animals-13-02756-t001:** Means and standard error (s.e.) for the color of subcutaneous fat depending on the type of animal. *p*-Values (significance of the animal type effect) for the different variables.

	Males	Females	Ewes	s.e.	*p*-Values
L*	63.4 b	65.2 b	70.4 a	0.795	<0.001
a*	4.6	3.9	3.8	0.330	0.577
b*	15.1	14.1	14.5	0.302	0.429
*h* _ab_	73.5	74.7	76.0	0.896	0.560
*C**_ab_	15.8	14.7	15.1	0.375	0.465

a, b—Differences among means (*p* < 0.05).

**Table 2 animals-13-02756-t002:** Means and standard error (s.e.) for shoulder weight (g) and tissue composition (percentage over shoulder weight) depending on the type of animal. *p*-Values (significance of animal type effect) for the different variables.

	Males	Females	Ewes	s.e.	*p*-Values
Shoulder weight (g)	1163.95 b	1074.88 b	2425.17 a	191.19	<0.001
Muscle (%)	58.50 a	52.94 b	57.85 a	0.91	0.007
Fat (%)	16.52	21.38	20.94	1.14	0.160
Bones and others (%)	23.74	24.42	20.11	0.85	0.072

a, b—Differences among means (*p* < 0.05).

**Table 3 animals-13-02756-t003:** Means and standard error (s.e.) for the pH and proximal composition of the meat depending on the type of animal. *p*-Values (significance of animal type effect) for the different variables.

	Males	Females	Ewes	s.e.	*p*-Values
pH	5.70	5.74	5.68	0.048	0.650
Moisture (%)	75.69 a	75.51 a	71.75 b	0.310	<0.001
Protein content (%)	22.09	22.15	22.16	0.151	0.932
Collagen content (%)	0.901 b	0.776 b	2.244 a	0.078	<0.001
Intramuscular fat content (%)	3.38 b	3.59 b	6.89 a	0.312	<0.001
Saturated fat content (%)	0.95 b	1.09 b	2.29 a	0.150	<0.001
Ash content (%)	2.15 b	2.04 b	3.39 a	0.06	<0.001

a, b—Differences among means (*p* < 0.05).

**Table 4 animals-13-02756-t004:** Means (mg/100 g) and standard error (s.e.) for the mineral content of meat depending on the type of animal. *p*-Values (significance of the animal type effect) for the different variables.

	Males	Females	Ewes	s.e.	*p*-Values
Ca	6.1 b	6.3 b	11.5 a	0.24	<0.001
Fe	1.9 b	1.9 b	3.3 a	0.36	<0.001
K	413.4 a	404.6 a	345.9 b	0.02	<0.001
Mg	28.0 a	28.7 a	25.3 b	0.06	<0.001
Na	62.1 a	60.5 a	46.9 b	0.05	<0.001
P	233.0 a	237.8 a	197.0 b	0.02	<0.001
Zn	2.6	2.6	2.8	0.23	0.216

a, b—Differences among means (*p* < 0.05).

**Table 5 animals-13-02756-t005:** Means and standard error (s.e.) for tocopherol, retinol, lutein, and cholesterol content depending on the type of animal. *p*-Values (significance of the animal type effect) for the different variables.

	Males	Females	Ewes	s.e.	*p*-Values
α-Tocopherol (µg/g fresh matter)	0.64 b	0.90 b	4.80 a	0.367	<0.001
γ-Tocopherol (µg/g fresh matter)	0.10	0.14	0.09	0.009	0.105
δ-Tocopherol (µg/g fresh matter)	0.014 b	0.020 a	0.016 b	0.001	<0.001
Retinol (µg/g fresh matter)	0.038 b	0.047 b	0.060 a	0.003	0.002
Lutein (ng/g fresh matter)	n.d.	0.531 b	6.490 a	0.977	0.006
Cholesterol (mg/g fresh matter)	0.680 b	0.681 b	0.769 a	0.012	0.001

a, b—Differences among means (*p* < 0.05). n.d., Not detectable.

**Table 6 animals-13-02756-t006:** Means (g/in 100 g of methylated fatty acids) and standard error (s.e.) for fatty acids depending on the type of animal. *p*-Values (significance of the animal type effect) for the different variables.

	Males	Females	Ewes	s.e.	*p*-Values
C8	0.035 b	0.048 ab	0.065 a	0.007	0.021
C10	0.158	0.150	0.167	0.015	0.702
C11	0.034 b	0.020 b	0.056 a	0.006	0.0005
C12	0.450 b	0.450 b	1.382 a	0.090	<0.0001
C13	0.547	0.046	0.061	0.004	0.075
C14	2.815	2.934	2.459	0.172	0.147
C14:1	0.255	0.241	0.255	0.020	0.833
C15	0.523	0.524	0.466	0.035	0.427
C15:1	0.109	0.117	0.147	0.011	0.067
C16	22.985	23.446	23.399	0.389	0.658
C16:1	2.027	2.153	2.038	0.096	0.600
C17:0	1.400 a	1.261 a	0.845 b	0.047	<0.001
C17:1	0.718	0.788	0.784	0.375	0.347
C18	12.620 a	13.217 ab	13.863 b	0.341	0.051
C18:1n9t	0.533 ab	0.548 a	0.381 b	0.050	0.048
C18:1n11t	0.136	0.090	0.116	0.016	0.153
C18:1n9c	44.23 a	42.988 b	43.246 ab	0.331	0.032
C18:2n6t	0.1411	0.112	0.147	0.11	0.063
C18:2n6c	7.628 a	7.826 a	5.755 b	0.478	0.009
C18:3n6g	0.127	0.104	0.084	0.017	0.224
C18:3n3a	0.329 b	0.378 b	0.954 a	0.044	<0.001
C20	0.115	0.096	0.114	0.0107	0.382
C22	0.027 b	0.026 b	0.067 a	0.004	<0.001
C20:1n9	0.078 b	0.065 b	0.128 a	0.009	<0.001
9c-11tCLA	0.357 b	0.377 b	0.577 a	0.038	<0.001
9c-11cCLA	0.093	0.063	0.088	0.012	0.177
10t12cCLA	0.084	0.057	0.082	0.012	0.232
C21	0.020 b	0.013 b	0.048 a	0.004	<0.001
C20:2	0.116 b	0.116 b	0.196 a	0.015	<0.001
C20:3n6	0.074 b	0.068 b	0.136 a	0.011	<0.001
C22	0.027 b	0.026 b	0.067 a	0.004	<0.001
C20:3n3	0.341	0.294	0.347	0.033	0.470
C23	0.031 b	0.030 b	0.066 a	0.007	0.002
C22:1	0.035 b	0.031 b	0.065 a	0.005	<0.001
C20:4n6n	0.539 a	0.445 ab	0.433 b	0.029	0.027
C22:2	0.010 b	0.010 b	0.034 a	0.002	<0.001
C24:0	0.010 b	0.010 b	0.034 a	0.002	<0.001
C20:5n3(EPA)	0.300	0.351	0.339	0.032	0.503
C24:1	0.010 b	0.010 b	0.034 a	0.002	<0.001
DPA	0.318	0.356	0.337	0.033	0.827
C22:6n3(DHA)	0.131 b	0.150 ab	0.199 a	0.015	0.010
Total n6	8.44 a	8.49 a	6.42 b	0.320	0.007
Total n3	1.18 b	1.24 b	1.98 a	0.082	<0.001
Ratio n6/n3	7.40 a	7.07 a	3.28 b	0.414	<0.001
Total SFA	41.3	42.3	43.1	0.54	0.075
Total MUFA	48.1	47.0	47.2	0.33	0.054
Total PUFA	10.59	10.70	9.71	0.54	0.380

a, b—Differences among batches (*p* < 0.05). SFA—saturated fatty acid; MUFA—monounsaturated fatty acid; PUFA—polyunsaturated fatty acid.

**Table 7 animals-13-02756-t007:** Means and standard error (s.e.) for texture variables depending on the type of animal. *p*-Values (significance of the animal type effect) for the different variables.

	Males	Females	Ewes	s.e.	*p*-Values
Hardness (N)	50.8	51.1	61.2	0.08	0.321
Adhesiveness (MJ)	0.22	0.19	0.22	2.30	0.937
Cohesiveness (dimensionless)	0.24 ab	0.27 a	0.22 b	0.01	0.004
Springiness (dimensionless)	2.48 b	2.70 ab	2.81 a	0.05	0.020

a, b—Differences among batches (*p* < 0.05).

**Table 8 animals-13-02756-t008:** Means and standard error (s.e.) for muscle color depending on the type of animal. *p*-Values (significance of the animal type effect) for the different variables.

	Males	Females	Ewes	s.e.	*p*-Values
L*	44.04 b	41.39 b	37.15 a	0.93	<0.0001
a*	6.20 b	5.82 b	10.33 a	0.76	<0.0001
b*	8.51	8.12	8.21	1.07	0.960
*C**_ab_	10.60	10.10	13.4	1.22	0.130
*h* _ab_	52.47 b	53.35 b	35.94 a	3.20	<0.0001

a, b—Differences among means (*p* < 0.05).

## Data Availability

The data presented in this study are available on request from the corresponding author.

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
