# Peer review of "Nutritional Quality of Meat from Barren Merino Ewes in Comparison to Meat from Traditional Lambs"

_animals, 2023, doi:10.3390/ani13172756_

Round 1

Reviewer 1 Report

The manuscript is important in terms of providing valuable information about “Nutritional quality of the Merino barren ewes’ meat in comparison to traditional light lambs’ meat”. I made some important recommendations for improving the proposed paper. 

1.      The manuscript contains some syntax errors and misspellings. Revise the text to improve readability. Linguistic revision is strongly suggested.

2.      In lines 16-17: change (males, females and males) to (males, females and ewes)

3.      Introduction; This introduction is really long and, in some cases, external to what you need to cover, so try to rephrase it by focusing on an introduction that is necessary for your study.

4.      Abbreviations should be explained when first used.

5.      Did you only feed the lambs? Didn't you feed the sheep? In the introduction, you talk about the important effect of feeding. How will you know whether the results you obtained in the study may have resulted from different feeding practices?

6.      In Table 3: When the table numbers are examined, it is seen that they start from 3. The table numbers all need to be rewritten.

7.      In line 232: Recheck the references given in the article and edit them according to the journal rules.

8.      In lines 221-247, 296-300: It is emphasized that the difference in results may be caused by feeding. This situation makes your differences controversial. How can we compare animals with different diets?

9.      In lines 257-258: What is meant in this sentence?

10.  In line 377: change (males, females and males) to (males, females and ewes)

11.  Too many references are used in the article. You can use the ones related to the subject and remove the others.

1.      The manuscript contains some syntax errors and misspellings. Revise the text to improve readability. Linguistic revision is strongly suggested.

Author Response

The manuscript is important in terms of providing valuable information about “Nutritional quality of the Merino barren ewes’ meat in comparison to traditional light lambs’ meat”. I made some important recommendations for improving the proposed paper. 

  1. The manuscript contains some syntax errors and misspellings. Revise the text to improve readability. Linguistic revision is strongly suggested.

We appreciate your comment. We have sent the manuscript to a reviewer to refine the style

  1. In lines 16-17: change (males, females and males) to (males, females and ewes)

Done

  1. Introduction; This introduction is really long and, in some cases, external to what you need to cover, so try to rephrase it by focusing on an introduction that is necessary for your study.

With all due respect, we consider that 63 lines cannot be considered as a long introduction, especially considering that the introduction is important to focus the reader on the problem to be solved, the state of the question and the background of the experiment carried out. However, with the aim of making it easier to read, we have eliminated the phrase “In fact […]

Abbreviations should be explained when first used.

We are totally agreed. Please, tell us which abbreviation you are referring to in order to correct it

  1. Did you only feed the lambs? Didn't you feed the sheep? In the introduction, you talk about the important effect of feeding. How will you know whether the results you obtained in the study may have resulted from different feeding practices?

With all due respect, we misunderstand this comment. In the introduction we do not talk at any time about the importance of feeding animals. What we are comparing are different commercial products (lambs or ewes) and we point out the importance of the sex and age of the animals in the characteristics of their meat, but we do not talk about the effect of the animals' diet. What we do want to point out is the nutritional interest (for people) of the different types of sheep meat.

Apart from this, we have not handled the animals. The animals have been raised on real farms, they come from different commercial farms, they are not an experimental herd. This means that what we are comparing are the products that these farmers, under the protection of the Extremadura lamb IGP, put on the market. Precisely for this reason, the introduction seems important to us, so that the context of the experiment be clear. Obviously, in the material and methods section we have to describe how the animals were raised, but they are not an experimental herd, we insist.

  1. In Table 3: When the table numbers are examined, it is seen that they start from 3. The table numbers all need to be rewritten.

We sincerely appreciate this comment. We have corrected the number of all tables and figures

  1. In line 232: Recheck the references given in the article and edit them according to the journal rules.

The text is written using the Endnote template for the Journal.

  1. In lines 221-247, 296-300: It is emphasized that the difference in results may be caused by feeding. This situation makes your differences controversial. How can we compare animals with different diets?

With all due respect, we still don't understand why the reviewer understood that we based the results on diet. At no time are we justifying our results based on the diet of the animals. In the titles of the tables, we clearly indicate that we are comparing "types of animals" and in the text we point out that the differences are due to age (ewes vs. lambs) or sex (males/females) but we do not talk about diet. When we make references to diet, we mean humans.

  1. In lines 257-258: What is meant in this sentence?

We would appreciate it if you could tell us exactly what concept or term is not understood.

  1. In line 377: change (males, females and males) to (males, females and ewes)

Done

  1. Too many references are used in the article. You can use the ones related to the subject and remove the others.

We have a total of 54 references. In general, journals’ editors recommend less than 60 and, in general, most of papers have around 50 references. We have re-checked the relevance of the references and we think that all of them are justified. 

Reviewer 2 Report

The Authors use Traditional light lambs in the title. Sanudo et al. (1998, Meat Sci, 49, S29-S64) defined light the carcasses  under 11 kg, others research wrote about light lamb: 12 to 16 kg, slaughtered at 45 and 60 days of age;  or  light lamb  9.5 to 12 kg.

In this experiment the carcass weight was 22-25 kg and the Authors in the Introduction specified that P.G.I.  ‘cordero de estremadura’ is 16 kg for males and 14 for females. 

I suggest to eliminate the adjective light from the title. 

In Material and methods as well as in Results paragraph is necessary careful revision because there are  many inaccuracies. Specify whether the analyses were done in duplicate or triplicate.

The following

Lines 16-17: There was no effect of animal type (males, females, and males) on pH etc.  correct: Female, male, female

Line 22: not saturated fat, but saturated fatty acids 

Line 58: substitute a study with  this  study

Lines 80-81: all components are on dry matter?  Specify

Line 82and 83: https://www.sciencedirect.com/science/article/pii/S092144881300148X#sec0005  light lamb: 12 to 16 kg, slaughtered at 45 and 60 days of age; https://www.sciencedirect.com/science/article/pii/S0309174000000413  light lamb  9.5 to 12 kg; 

Line 83 kg small letter and not Kg capital letter 

Lines 90-91: please, control Minolta 600d or Minolta 2006d?  (repeated twice)

Line 94: 90 degrees? WHY?

Line 97: chroma Cab  

Lines 104-105: Panea et al. 

Line 107: triceps brachii or triceps brachialis muscle

Line 109: Specify when  the pH was measured  

Line 127: UPLC (lines 123,124, 126) or UHPLC?

Lines 129 to 133: missing verb in sentence 

Line 136: as line 107 and use italics for Latin name of muscle

Line 138: Why qualitative percentage? 

Line 141: 2 as subscript 

Line 150: patterns? what are they? standard FAME? 

Line 164: fat colour of the fat, delete of the fat. The first table, is Table 1 and not  Table 3. Also applies to Tables to follow  

Line 175-177: The table header could be written better. Also applies to tables to follow. 

p<0.05 and not p<0,05

Line 200 (in the table): Why saturated fat content? shouldn't it be saturated fatty acids?  

Lines 210-211: Baila et al  

Line 213: mg per g of meat? 

Lines from 215 to 220: Saturated fat? Saturated fatty acids

Line 217: should be? must be? have to be? 

Line 232: Novoselec et al 

Line 238: Holman et al (33)

Line 261: Alvarez et al 

Line 263: Yang et al  

Line 266: Alvarez et al 

Line 274-275: the significance of comparisons is ewes (females) vs lambs (males and females).

Lines 278-279: Campo et al  

Line 280: g of meat? Of muscle only? Of fat only? Please, specify 

Line 285 (unit of measurement in the table): grams of what? obviously applies to all parameters 

Lines 290-291-292: The higher content of individual fatty acids of the ewes is justified by the Authors with the higher fat content, but the FAs in the table are given as a percentage of total fatty acids and so in my opinion this higher content of individual FAs is not related to the higher overall fat content but to other factors, perhaps diet. In fact what is different is the fatty acids profile and not the amounts in absolute sense. 

Lines 329: is the first figure, why Figure 2? 

Lines 330-331: no explanation why, since the one with the most fat is sheep and not lambs, so in theory more prone to oxidation. 

Lines 351-352: Martinez-Cerezo et al 

Line 367: iron atoms? Better molecule; ewes not males

Author Response

The Authors use Traditional light lambs in the title. Sanudo et al. (1998, Meat Sci, 49, S29-S64) defined light the carcasses  under 11 kg, others research wrote about light lamb: 12 to 16 kg, slaughtered at 45 and 60 days of age;  or  light lamb  9.5 to 12 kg.

In this experiment the carcass weight was 22-25 kg and the Authors in the Introduction specified that P.G.I.  ‘cordero de estremadura’ is 16 kg for males and 14 for females. 

I suggest to eliminate the adjective light from the title. 

- We sincerely appreciate your comment. We have removed the word light from the article.

In Material and methods as well as in Results paragraph is necessary careful revision because there are  many inaccuracies. Specify whether the analyses were done in duplicate or triplicate.

- done

The following

Lines 16-17: There was no effect of animal type (males, females, and males) on pH etc.  correct: Female, male, female

- done

Line 22: not saturated fat, but saturated fatty acids 

- done

Line 58: substitute a study with  this  study

- done

Lines 80-81: all components are on dry matter?  Specify

Line 82and 83: https://www.sciencedirect.com/science/article/pii/S092144881300148X#sec0005  light lamb: 12 to 16 kg, slaughtered at 45 and 60 days of age; https://www.sciencedirect.com/science/article/pii/S0309174000000413  light lamb  9.5 to 12 kg; 

- We do not fully understand this comment. We assume that you are referring to the previous comment about the adjective "light". As we have commented, we have eliminated the adjective.

Line 83 kg small letter and not Kg capital letter 

- done

Lines 90-91: please, control Minolta 600d or Minolta 2006d?  (repeated twice)

- Thank you for the comment. Corrected.

Line 94: 90 degrees? WHY?

Thank you for the comment. It was a typing mistake. Corrected.

Line 97: chroma Cab  

Corrected

Lines 104-105: Panea et al. 

We do not understand what this comment refers to. The reference is from a dissection atlas published in the ITEA journal. If you mean how the citation appears, the manuscript is written using the Endnote template for the journal

Line 107: triceps brachii or triceps brachialis muscle

We sincerely appreciate this correction. It was a mistake; we have already changed it.

Line 109: Specify when  the pH was measured 

done

Line 127: UPLC (lines 123,124, 126) or UHPLC?

Lines 129 to 133: missing verb in sentence 

We have reviewed the paragraph and we have not seen that the verb is missing. Please tell us which verb is missing.

Line 136: as line 107 and use italics for Latin name of muscle

We sincerely appreciate this correction. It was a mistake; we have already changed it.

Line 138: Why qualitative percentage? 

What we wanted to say is that the method allows quantification but also qualitative identification (which acids have been detected). However, we have removed qualitative to make it clearer.

Line 141: 2 as subscript 

Done

Line 150: patterns? what are they? standard FAME? 

Yes, FAMES standard, patterns to identify which fatty acids appear in the analysis.

Line 164: fat colour of the fat, delete of the fat.

done

The first table, is Table 1 and not  Table

done

  1. Also applies to Tables to follow  

corrected

Line 175-177: The table header could be written better. Also applies to tables to follow. 

Corrected

p<0.05 and not p<0,05

done

Line 200 (in the table): Why saturated fat content? shouldn't it be saturated fatty acids?  

The NIRS Foss that we use to determine the proximal composition gives the results as percentage of saturated fat. Obviously, they will be saturated fatty acids, but we prefer to put the notation as given by the software of the machine.

Lines 210-211: Baila et al  

We assume that it refers, as in Panea et al, to the format. We refer you to our previous comment.

Line 213: mg per g of meat? 

Yes. Corrected

Lines from 215 to 220: Saturated fat? Saturated fatty acids

As we said in line 200, what the NIRS gives is the percentage of saturated fat and the EFSA recommendations are for saturated fat, so we prefer to use this term which, on the other hand, is easier to understand for the non-expert reader.

Line 217: should be? must be? have to be? 

Should, they are recommendations. We changed the sentence.

Line 232: Novoselec et al 

We refer you to our previous comment.

Line 238: Holman et al (33)

We refer you to our previous comment.

Line 261: Alvarez et al 

We refer you to our previous comment.

Line 263: Yang et al  

We refer you to our previous comment.

Line 266: Alvarez et al 

We refer you to our previous comment.

Line 274-275: the significance of comparisons is ewes (females) vs lambs (males and

females).

Changed, thank you very much, again.

Lines 278-279: Campo et al  

We refer you to our previous comment.

Line 280: g of meat? Of muscle only? Of fat only? Please, specify 

Results are /100 g of edible portion, which is, following the cited authors, muscle + visible fat. We indicated it into the text.

Line 285 (unit of measurement in the table): grams of what? obviously applies to all

parameters 

Corrected

Lines 290-291-292: The higher content of individual fatty acids of the ewes is justified by the Authors with the higher fat content, but the FAs in the table are given as a percentage of total fatty acids and so in my opinion this higher content of individual FAs is not related to the higher overall fat content but to other factors, perhaps diet. In fact what is different is the fatty acids profile and not the amounts in absolute sense. 

As the reviewer knows, total amount of fat influences the fatty acid composition. Although some studies show the absolute values ​​to give a better idea of ​​the nutritional composition, most of the bibliography continues to express them as a percentage (or g/100g, which is how we express it). Knowing the fat content of the animal, it is easy to calculate the content of each fatty acid.  Obviously, the diet of the different types of animals used will influence the fatty acid profile (just as influence the amount of fat) and for this reason we describe in material and methods the handling of each group of animals, and an indication is made on line 338 when speaking about the CLA content.  However, the present study is not a comparison of diets (over which we have no control), but of commercial products. Therefore, we focus on it. On the other hand, the influence of age is undeniable.   What we want to highlight is that, despite the fact that the meat of adult animals often has a bad reputation because an inadvisable lipid profile is attributed to them, our data have shown that, beyond the fact that there may be differences in any specific acid, there are no differences in the sums, so the meat of these sheep can be considered healthy. We have included some sentences to highlight this reasoning. We hope they add to clarify the reading.

Lines 329: is the first figure, why Figure 2? 

We very appreciate this comment. We corrected all the tables and figure numbers.

Lines 330-331: no explanation why, since the one with the most fat is sheep and not lambs, so in theory more prone to oxidation. 

We appreciate this comment because it is true that it is an unexpected result, considering that there are no differences in the percentage of unsaturated fat. we included a phrase in this sense.

Lines 351-352: Martinez-Cerezo et al 

We refer you to our previous comment.

Line 367: iron atoms? Better molecule; ewes not males

With all due respect, as the sure reviewer knows, myoglobin has a heme prosthetic group with an iron atom at its centre.

Changed males by ewes.

Round 2

Reviewer 1 Report

Dear Editor,

It appears that the authors have revised the article in line with our revisions. But there are a few minor fixes. These are below; In line 48, Correct "famers" to "farmers"

In line 389, Table 6 or Table 4? Iron contents are given in Table 4. Check and fix it.

Reviewer 2 Report

The paper was improved by the Authors that accepted suggestions. I agree with its publication in Animals